# The Relationship between Emotional Intelligence and Pain Management Awareness among Nurses

**DOI:** 10.3390/healthcare10061047

**Published:** 2022-06-04

**Authors:** Marwan Rasmi Issa, Noor Awanis Muslim, Raed Hussam Alzoubi, Mu’taman Jarrar, Modhi A. Alkahtani, Mohammad Al-Bsheish, Arwa Alumran, Ammar K. Alomran

**Affiliations:** 1Skills Development Training Center, King Saud Medical City, Riyadh 12746, Saudi Arabia; m.alkahtani@ksmc.med.sa; 2College of Graduate Studies (COGS), Universiti Tenaga Nasional (UNITEN), Putrajaya 43000, Malaysia; awanis@uniten.edu.my; 3Department of Administrative Sciences, Prince Hussein Bin Abdullah Academy for Civil Protection, AL Balqa Applied University, As-Salt 19117, Jordan; Raed.Alzoubi@pha.edu.jo; 4Vice Deanship for Quality and Development, College of Medicine, Imam Abdulrahman Bin Faisal University, Dammam 34212, Saudi Arabia; mkjarrar@iau.edu.sa; 5Medical Education Department, King Fahd Hospital of the University, Al-Khobar 34445, Saudi Arabia; 6Health Management Department, Batterjee Medical College, Jeddah 21442, Saudi Arabia; mohammed.ghandour@bmc.edu.sa; 7Health Information Management and Technology, College of Public Health, Imam Abdulrahman Bin Faisal University, Dammam 34212, Saudi Arabia; aalumran@iau.edu.sa; 8Department of Orthopedic, College of Medicine, Imam Abdulrahman Bin Faisal University, Dammam 31441, Saudi Arabia; aomran@iau.edu.sa

**Keywords:** pain management awareness, emotional intelligence

## Abstract

**Background:** Pain management, a crucial part of nursing care, is considered one of the most basic patient rights. To properly treat patients’ pain, nurses need a high degree of pain management awareness (PMA). The researchers hypothesized that nurses’ pain management awareness is affected by their emotional intelligence (EI). **Purpose:** Because there is a dearth of studies on this topic, the purpose of this study was to describe the relationship between emotional intelligence and pain management awareness in a sample of nurses. **Methods:** The study employed a descriptive design with a quantitative approach to analyze data from a survey designed with the simple random sample technique. The questionnaires were completed by 330 nurses working at six governmental hospitals in Saudi Arabia. The Statistical Package for the Social Sciences (V23) and Analysis of Moment Structures (V23) were used to determine the reliability and validity of the questionnaires and analyze the causal relationships among the variables. **Results:** The results revealed a significant positive relationship between nurses’ emotional intelligence and their pain management awareness. **Conclusions:** These findings suggest that having emotional intelligence is an important nurse characteristic for effective pain management awareness and possibly the provision of pain management care. **Clinical Implications:** Hospital and nurse managers as well as administration should consider using the emotional intelligence variables utilized in this study to develop ways to improve pain management awareness among nurses. Such efforts may help improve hospital and patient outcomes related to pain management.

## 1. Background

Pain management is one the core functions of the healthcare sector and an essential component of nursing care. As one of the most basic patient rights, nurses have an ethical obligation to provide their patients with effective pain management care. Pain management is important for all patients, considering that it affects their lifespan and requires special skills and care. It also differs from one patient to another as pain management for older people is more demanding compared to adults. Elderly patients face some cognitive impairments and require more treatments [1].

The process of pain management comprises three stages. The first phase involves the assessment of patients’ pain with the appropriate pain assessment instrument, followed by the second phase that involves treating the patients’ pain with pharmacological or non-pharmacological intervention, and the last phase where the patients’ pain is reassessed [2]. Relaxation, a cognitive strategy that incorporates the distraction method, reduces the severity of pain and improves the patients’ ability to control pain [3].

The studies exploring the relationship between nurses’ emotional intelligence (EI) and pain management awareness (PMA) were inadequate. Emotional intelligence plays a vital role in the healthcare sector, as indicated by the strong relationship between emotional intelligence and the healthcare-related field according to the healthcare literature. A study was conducted on the relationships between nursing students’ perceptions of hospice and palliative care and their EI. A total of 458 nursing students were involved in the study. As a result, it was found that the perceptions of hospice and palliative care were significantly and positively correlated with EI, which should be supported to improve nurses’ perceptions of hospice and palliative care and incorporated into related curricula and extracurricular programs [4].

A study in the U.S. regarding the use of EI as the educational strategy for teaching safety in nursing practice demonstrated a positive relationship between EI and safety in ineffective nursing practice [5]. Notably, EI is one of the most interesting areas for improvement in nurses’ performance. Based on a cross-sectional study conducted in the U.S. among university students to examine the EI between the 113 first-year nursing students and 104 engineering students, it was found that nursing students had higher levels of EI compared to the engineering students, with the female students having higher EI compared to male students. The Trait Emotional Intelligence Questionnaire (TEIQue) and the Schutte Self Report Emotional Intelligence Test (SSEIT) were employed in the study [6], although no correlation was present between emotional intelligence and clinical teaching among New York state nursing students [7].

A study was conducted to determine whether improved functional outcomes (physical and psychosocial daily functioning) and increased self-efficacy are connected with mindfulness, psychological flexibility, and emotional intelligence among people with chronic pain. Emotional intelligence was found to be significantly associated with self-efficacy among people with chronic pain [8]. In another study, an evaluation was conducted on how nurses learn caring behaviors in nursing school and apply the behaviors in the hospital. As a result, patients, families, and colleagues who were treated by nurses with emotional intelligence would gain a more satisfying experience, allowing both nurses and patients to gain positive healthcare experiences. Furthermore, nurses who deliberately developed EI would allow their colleagues and patients to succeed in their healthcare experiences [9]. Based on a study conducted in Alabama and Long Island, New York in the United States, one of the most notable findings was recorded about the relationship between emotional intelligence and the ability to repair the moods of the patients suffering from pain [10] among adult patients in different healthcare centers.

Although another research work was conducted by Richard Mendelson (2019) about the relationship between emotional intelligence and control over pain from urology problems, a strong correlation was present for the high patient level of emotional intelligence with low level of pain [11]. However, a strong relationship was recorded between emotional intelligence and patients’ central care. The Provider–Patient Relationship Questionnaire was employed in a study among 318 Italian healthcare workers. It was found that emotional intelligence directly affected patient care, indicating that emotional intelligence could enhance the quality of care in healthcare facilities [12]. Nurses had a low level of perception regarding pain management in the emergency department of one of the public hospitals in South Australia. To illustrate, the finding in the qualitative study showed that they were initially omitted from the patient diagnosis process [13].

The evaluation was conducted on the relationships between self-efficacy for pain management and experience of pain and the relationship between emotional intelligence and understanding of pain among patients. Furthermore, the research investigated whether self-efficacy for pain management and emotional intelligence functioned as the mediator of the relation between mindfulness and pain experience. Increased self-efficacy in pain management and emotional intelligence were associated with decreased subjective pain experience. Following that, emotional intelligence and self-efficacy in managing pain were found to be significant mediators of the connection between mindfulness and pain [14]. A study was conducted among nursing students in Hong Kong to elaborate on the effects of emotional intelligence, knowledge, and attitude on pain management. As a result, these variables were significant in pain assessment and management, while the negative relationship was substantial although insignificant among first-year students. The result strongly suggested for the nurse leader to maintain emotional intelligence and for the nurses to improve the process of patient care as an essential daily practice work [15]. Therefore, further investigation on the correlation is required [16]. Meanwhile, research was conducted in 11 small and medium hospitals in Korea regarding the effect of emotional intelligence and work environment on nursing performance, which was mediated by communication competence. This study involved 240 nurses. It was found that communication competence mediated the relationships of emotional intelligence with the work environment on the nurses’ performances [17]. Another study was conducted in Spain to evaluate the emotional intelligence among nursing students after the educational intervention. A total of 103 students were involved in the pre- and post-test study and went through emotional and educational interventions in different workshops. It was found that the nursing students’ emotional intelligence levels increased [18].

Although the issue of nurse emotional intelligence had been examined in previous studies, inadequate emphasis was placed on emotional intelligence as it could form a barrier to the improvement in pain management awareness among nurses. Emotional intelligence plays a vital role in enhancing different healthcare aspects, which creates critical importance for emotional intelligence to be involved in pain management processes. The available literature did not delve into nurses’ emotional intelligence to explore the flaws in pain management awareness among nurses and the method of improving hospitals systems in this matter.

### Purpose

The present research hypothesized that nurses’ PMA is affected by EI. Because there is a dearth of studies on this topic, this study aimed to examine the relationship between EI and PMA in a sample of nurses.

## 2. Definitions of Variables

### 2.1. Pain Management Awareness

The areas of pain management self-efficacy that require development to increase the nurses’ understandings of issues in pain management were identified. This process is important for safe and quality patient care. It could also be used for ground quality improvement and professional enhancement strategies. This management begins with the assessment of pain and ends with pain control from all healthcare workers [19].

### 2.2. Emotional Intelligence

Emotional intelligence denotes the proficiency of people to perceive their own emotions and those of others. It determines various feelings and labels them accordingly [20]. 

### 2.3. Social Emotion

Social emotion relates to our self-awareness and our perception of others’ reactions to us.

### 2.4. Use of Emotion

It is in the susceptible space of emotional communications that we link through empathy. 

### 2.5. Regulation of Emotion

It is the ability to respond to the ongoing demands of experience with a variety of emotions in a socially appropriate and satisfactorily versatile manner that allows for spontaneous reactions as well as the ability to postpone spontaneous reactions as required.

## 3. Research Framework

In this research, the conceptual framework seeks to test the relationship between emotional intelligence and pain management awareness among nurses and explain the relationships of some variables with pain management awareness. The central concept behind the model is pain management awareness which is formed by all factors in emotional intelligence. This research model is slightly more comprehensive with more variables. Pain management awareness is the dependent variable, and the independent variables for this study were emotional intelligence (self-emotional appraisal (SEA), others’ emotional appraisal (OEA), regulation of emotion (ROE), and use of emotion (UOE)). This research model introduces one new variable, which is emotional intelligence (Figure 1).

## 4. Theoretical Framework

### 4.1. Behavioral Change Theories

Any effective strategy to improve nurses’ PMA levels must be based on the understanding of health behaviors and the situations in which they develop. Therefore, strategies designed to affect any health behavior must be constructed using behavioral change theories. A literature search revealed a dearth of studies using quasi-experimental designs for studying the effects of educational strategies on nursing staff behaviors. Unfortunately, little is known about whether educational strategies can affect the nursing staff members’ self-efficacy. A theory can be prescribed as “a group of complementary perception, interpretation, and the hypothesis that perform an organized framework for the situations by determining the relationship between variables to anticipate events or situations” [21]. Based on this prescription, any health behavioral theory should describe the pertinent variables and analyze how these variables interact [22].

### 4.2. EI Theory

This study was influenced by Daniel Goleman’s EI theory [23], which defines EI as acquiring self-awareness, self-regulation, motivation, empathy, and social skills. At the time when it was developed, the theory of EI was progressive because it assumed that knowledge alone is insufficient to achieve goals. Instead, EI is how people know, act logically, and connect with others via their emotions, perceptions, and feelings. The four components of the EI model are social awareness, self-awareness, relationship management, and self-management. Furthermore, to have EI, individuals should not only be solely self-oriented but also be aware of others and social relationships. For example, when nurses are more self-oriented about pain management, they can develop ways to be more patient-oriented regarding pain management. EI is divided into four competencies: describing emotions nonverbally, using emotions to manage cognitive thought, understanding the knowledge that emotions transfer and the behaviors that emotions produce, and regulating one’s own emotions for personal and societal gain.

EI can also promote the best possible treatment for the patient; when nurses do not use EI while interacting with patients, their treatment may be limited. For example, if a nurse responds to a call from a patient who is in extreme pain and the nurse does not offer enough time to manage the patient’s pain and immediately leaves to handle another task, the patient may not feel important to the nurse. When this occurs, the nurse–patient relationship likely would be harmed. A nurse with a higher degree of EI will handle the patient correctly, allowing ample time to manage the patient’s pain. The more EI nurses have, the more self-aware they are of their feelings, which can promote the understanding of the patient’s emotions and how to cope with them. This pathway helps ensure that the patient is comfortable with the nursing treatment. The nurse has the time and expertise to treat pain, regardless of their training, thereby strengthening the nurse–patient clinical partnership. In this way, emotional self-awareness can contribute to better pain management.

## 5. Development Research Hypotheses 

Research hypothesis is described as an informed postulation, guess, or prediction formulated in line with the supposed relationships between two or more research constructs [24]. Thus, it follows that research hypothesis is defined as an educated speculation or forecast that considers the existence of a relationship between two or more research constructs [25]. According to Sekaran, U. and Bougie, a research hypothesis could be developed from the research conceptual framework by identifying the critical constructs [26]. The following hypotheses (H) were formulated to be tested using various statistical analyses.

**Hypothesis 1** **(H1).***Self-emotional appraisal* *has a positive effect on pain management awareness (PMA).*

**Hypothesis 2** **(H2).***Others’ emotional appraisal* *has a positive effect on pain management awareness (PMA).*

**Hypothesis 3** **(H3).***Use of emotion* *has a positive effect on pain management awareness (PMA).*

**Hypothesis 4** **(H4).**
*Regulation of emotion has a positive effect on pain management awareness (PMA).*


## 6. Materials and Methods

For this quantitative study, the researchers employed an empirical and descriptive design. Data were generated from questionnaires completed by 330 nurses who currently work with patients suffering from pain at six governmental hospitals in the Kingdom of Saudi Arabia. The study was approved by King Saud Medical City’s institutional review board and the Ministry of Health. All nurses provided informed consent before participating in this study.

The 16-question EI questionnaire used in this study was adopted from Wong and Law Emotional Intelligence Scale [27] and is scored using a five-point Likert scale. This questionnaire identifies four categories of EI: self-emotional appraisal (SEA), others’ emotional appraisal (OEA), regulation of emotion (ROE), and use of emotion (UOE). The questionnaire’s reliability calculated using Cronbach’s alpha was 0.894.

The PMA questionnaire was adopted from [28] and included 21 items concerning participants’ PMA level. The items were scored using a five-point Likert scale with higher scores indicating better self-assessed PMA levels. The questionnaire contained statements, such as rate the ability to “identify patient characteristics affecting pain management.” The scale’s reliability calculated using Cronbach’s alpha was 0.983.

After the data collection, a coding procedure was developed in which the data was coded and edited before being transferred to the computer for data analysis. The systems used for data analysis, (i) IBM SPSS (Statistical Package for the Social Sciences; version 23), were used to provide a wide range of quantitative statistical data, such as descriptive statistics, including the measure of central tendency (mean, median, mode) and the measurement of dispersion (range, standard deviation, percentiles). The analysis performed in this study included paired sample t-test, Pearson Product Moment Correlation Coefficient (r). Exploratory data analysis (EDA) was also conducted to examine the normality, linearity, and multicollinearity. (ii) Analysis of Moment Structures (AMOS) V23 software was used for confirmatory factor analysis or measurement model (CFA), while structural equation modelling (SEM) was employed to examine the goodness of fit for the proposed models and subsequently estimate the coefficients about the hypothesised path models.

### Sample Size

The population size in this study was 2200 nurses. Contrary to the aforementioned techniques, Krejcie and Morgan fulfilled these research criteria and covered the population for this research. Krejcie and Morgan (1970) highlighted that the sample size is 327 nurses [29]. However, this ratio is unachievable, considering that some nurses may choose to not participate or not return the questionnaire. This condition leads to a reduction in the sample size required. To avoid this issue, a larger sample size (N 420) was employed in this study. The Analysis of Moment Structure (AMOS) V23 was used for the data analysis in this study. 

## 7. Results

The following demographic information was gathered from the 330 respondents: gender, age, marital status, education level, job position, experience, attendance at pain education programs, and working area (Table 1). All respondents were staff nurses, and most were female (70%). Regarding age, 43.3% of the nurses were 25–34 years old, 33.3% were 35–44 years old, 13.3% were 45–54 years old, and 10% were <25 years old. Among the sample, 73.3% had a bachelor-level qualification.

The results for the alpha value reliability for PMA (dependent variable) and all independent variables (SEA, OEA, ROE, and UOE) are shown in Table 2. All dependent and independent variables had relatively strong reliability. The alpha value was 0.869 for UOE, 0.983 for PMA, 0.897 for ROE, 0.894 for SEA, and 0.875 for OEA.

The Pearson correlation matrix between the dependent variable and all independent variables are shown in Table 3. Among all variables, the correlation was significant at the 0.01 level (two-tailed), indicating a significant positive correlation between EI and PMA.

The model used in this study was fit according to the analysis of variance; **the** results showed a significant value of <0.001 (Table 4). This finding supports the research hypothesis that indicates that EI variables had a positive effect on PMA, and the model is fitted as a significant value of <0.05.

The results of the analysis of the direct effects of the constructs using analysis of moment structures indicated that EI had a positive effect on PMA (Table 5). The critical ratio (c.r.) and *p*-value of EI in predicting PMA were 5.353 and <0.001, respectively. This finding implies that the probability of getting a c.r. as large as 5.353 in absolute value is <0.001. In other words, the regression weight for EI in the prediction of PMA was significantly different from zero at the 0.001 level (two-tailed). Furthermore, the standardized estimate of beta was 0.314, indicating a positive relationship. In other words, when EI increases by one standard deviation, PMA increases by 0.314 standard deviations.

## 8. Discussion

The emotional intelligence factor is employed by nurses globally. However, the research explaining the relationship between emotional intelligence and pain management awareness among nurses is lacking. In this thesis, the EI was employed as a significant direct relationship with pain management awareness. This finding has developed a new contribution to the pain management field. In line with this, a study was conducted to determine the emotional intelligence among people with chronic pain, which was found to be significantly associated with self-efficacy among people with chronic pain [8]. In another study, an evaluation was conducted on how nurses learn caring behaviors in nursing school and apply the behaviors in the hospital area. Specifically, patients, families, and colleagues who are exposed to nurses and have gained emotional intelligence would gain a more satisfying experience, which allows both nurses and patients to thrive in their healthcare experiences. Furthermore, nurses who intentionally develop their emotional intelligence would allow their colleagues, patients, and colleagues to gain positive healthcare experiences [9]. As for emotional intelligence in the U.S. (2015), the educational strategy for teaching safety in nursing practice showed a positive relationship between emotional intelligence and safety in the nursing practice [5]. To improve nurse PMA, hospital and nurse managers and administration should consider developing a pain management education program to address the four EI components used in this study. Indeed, improving nurse PMA may contribute to improved hospital and patient outcomes.

The theory of emotional intelligence [23] is related to self-awareness, self-regulation, motivation, empathy, and social skills. It is progressive on the assumption that knowledge alone is not adequate for individuals to achieve the target. To increase pain management awareness, nurses should improve their self-awareness, social skills, empathy, and motivation. Based on the theory, emotional intelligence could be used as one of the main variables affecting pain management. It is also one of the most interesting areas for improving nurses’ performance. Moreover, one cross-sectional study was conducted in the U.S. to examine the emotional intelligence among 13 nursing students in the first year and 104 engineering students. As a result, the nursing students showed a higher level of emotional intelligence compared to the engineering students, while the emotional intelligence level among the female students was higher compared to the male students from both groups. The Trait Emotional Intelligence Questionnaire (TEIQue) and Schutte Self Report Emotional Intelligence Test (SSEIT) were employed in the study [6].

One more study supports this research findings, which is the qualitative phase of a larger mixed-method study to explore clinical nurses’ experiences of utilizing emotional intelligence capabilities during clinical reasoning and decision making. The result shows the sensibility to engage EI capabilities in clinical contexts; motivation to actively engage with emotions in clinical decision making; and incorporating emotional and technical perspectives in decision -making [30].

As previously mentioned, emotional intelligence affected the nurses’ performance and practice. Therefore, it was used with the nurses to improve their awareness of pain management. This study filled the current gaps in emotional intelligence towards pain management awareness among the participating nurses, which is different than the previous studies.

## 9. Practical Implications of the Study

The results of this research demonstrated the possibility to predict the improvement in pain management awareness based on the changes in EI and health beliefs among the nurses. This factor is a significant practical implication of this study, which creates a new opportunity to consider and offer new ideas on enhancing pain management depending on the understanding of predictor variables for this research. It was also demonstrated that the nurses’ beliefs and emotional intelligence about pain management were associated with awareness.

## 10. Limitations of the Study

This study specifically focused on the hospitals in Saudi Arabia. Therefore, caution was applied due to the fact that the research was conducted within a specific sector of governmental hospitals and in one country (Saudi Arabia). However, replications in other contexts would increase the confidence of the research model and offer other researchers an opportunity to conduct a comparative analysis with the results of the current study. Furthermore, this study was limited to nurses and did not include any other health occupational group (physicians, pharmacists, technicians). Therefore, the results could be generalized to staff nurses working in governmental hospitals. The methodology could be replicated in other areas and for different populations. Nevertheless, upon the focus on the technical aspects of pain management awareness among nurses and improvement in the pain management practice in the hospitals, several cost-cutting issues related to pain management among hospitals should be considered by the researchers.

Besides the deep insights into the impact of emotional intelligence on pain management awareness, several limitations were present in this study. This study has presented a framework to change nurses’ beliefs and behaviours by investigating the EI among nurses. However, covering all the possible variables was impossible, such as environmental factors, psychosocial factors, and nurses workload barriers to reporting pain, and nursing-patient ratio [31,32]. Future research may explore the nursing new learning strategy in helping the nurses improve pain management awareness [33]. Specifically, the use of quantitative data significantly reduced the scope and applicability of the research. Therefore, future researchers should focus on the use of appropriate research methods and complementing quantitative data with qualitative methods to strengthen the study findings and the confidence in the results. 

## 11. Conclusions

The results of this study supported this argument. Nurses ought to incorporate emotional intelligence instructional tactics as part of their pre-licensure nursing degrees and professional development programs. Besides, nurse supervisors should inspire the nurses to integrate emotional intelligence into daily practice to achieve comprehensive patient care.

This result tells us that those nurses with more emotional intelligence are effectively looking for better pain management than those with low levels of emotional intelligence; this finding is considered a new contribution to the pain management field. By taking a more systemic approach, this study was able to explore the validity of earlier findings and identify a new concept that has not been a focus of the empirical research conducted to develop a better understanding of them and examine their relationship.

## Figures and Tables

**Figure 1 healthcare-10-01047-f001:**
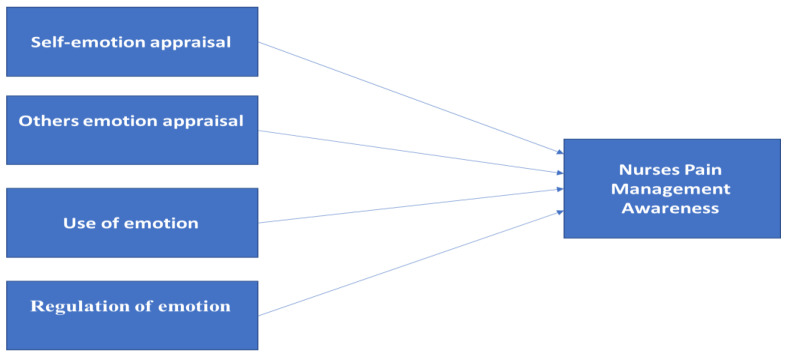
Research framework on the relationship between emotional intelligence and nurses pain management awareness.

**Table 1 healthcare-10-01047-t001:** Sample Profile.

Items	Frequency	Percentage
Gender		
Male	99	30.0
Female	231	70.0
Age		
Less than 25 years	33	10.0
25–34 years	143	43.3
35–44 years	110	33.3
45–54 years	44	13.3
Marital Status		
Single	88	26.7
Married	242	73.3
Educational Level		
diploma/high diploma	66	20.0
Bachelor	242	73.3
Master	22	6.7
Working Experience		
Less than 3 years	22	6.7
3–6 years	154	46.7
7–10 years	110	33.3
11–14 years	44	13.3
Area		
Medical Ward	55	16.7
Surgical Ward	44	13.3
Oncology Ward	55	16.7
Intensive Care Unit	77	23.3
Emergency Room	55	16.7
Operation Room	44	13.3

**Table 2 healthcare-10-01047-t002:** Reliability Analysis.

Variables	No. of Items	Cronbach Alpha	M	SD
Pain Management Awareness (PMA)	21	0.983	3.668	0.811
Self-Emotional Appraisal (SEA)	4	0.894	3.651	0.763
Others Emotional Appraisal (OEA)	4	0.875	3.630	0.777
Use of Emotion (UOE)	4	0.869	3.701	0.870
Regulation of Emotion (ROE)	4	0.897	3.585	0.826

Note: M—mean, SD—standard deviation.

**Table 3 healthcare-10-01047-t003:** Pearson Correlation Matrix.

Variables		SEA_Mean	OEA_Mean	UOE_Mean	ROE_Mean	PMA_Mean
SEA_mean	Pearson Correlation	1	0.663 **	0.666 **	0.691 **	0.251 **
	Sig. (2-tailed)		0.000	0.000	0.000	0.000
	N	330	330	330	330	330
OEA_mean	Pearson Correlation	0.663 **	1	0.691 **	0.658 **	0.216 **
	Sig. (2-tailed)	0.000		0.000	0.000	0.000
	N	330	330	330	330	330
UOE_mean	Pearson Correlation	0.666 **	0.691 **	1	0.556 **	0.247 **
	Sig. (2-tailed)	0.000	0.000		0.000	0.000
	N	330	330	330	330	330
ROE_mean	Pearson Correlation	0.691 **	0.658 **	0.556 **	1	0.182 **
	Sig. (2-tailed)	0.000	0.000	0.000		0.001
	N	330	330	330	330	330
PMA_mean	Pearson Correlation	0.251 **	0.216 **	0.247 **	0.182 **	1
	Sig. (2-tailed)	0.000	0.000	0.000	0.001	
	N	330	330	330	330	330

Note ** Correlation is significant at the 0.01 level (two-tailed).

**Table 4 healthcare-10-01047-t004:** ANOVA.

Model	Sum of Square	Df	Mean Square	F	Sig.
Regression	14.315	1	14.315	23.239	0.000
Residual	202.047	328	0.615		
Total	216.363	329			

Dependent variable: PMA_mean; Predictors (constant): EI_mean.

**Table 5 healthcare-10-01047-t005:** Examining Results of Direct Effects of the Constructs by (AMOS).

Path	Unstandardized Estimate		Standardised Estimate	Critical Ration (c.r.)	*p*-Value	Hypothesis Result
	Estimate	S.E.	Beta			
EI→PMA	0.327	0.061	0.314 ***	5.353	0.000	Supported

*** *p* < 0.001.

## Data Availability

The data supporting the findings of this study are available from the corresponding authors upon reasonable request.

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
