# Peer review of "The Relationship between Emotional Intelligence and Pain Management Awareness among Nurses"

_healthcare, 2022, doi:10.3390/healthcare10061047_

Round 1

Reviewer 1 Report

This work examines the association between pain management and emotional intelligence in nurses from Saudi Arabia. It is well-written but it requires some improvements:

a) Further literature review is needed on the relationship between pain managemente and EI. 

b) Hypotheses shouls be presented and contrasted.

c) A more thorough data analysis is recommeded, by conducting regression analyses with the different dimensions of EI and controlling for demographics. 

d) Further development is needed in the discussion, in order to explain the results and understand some possible implications for practice and policy in that context.

e) Limitations should be acknowledged in the discussion.

Author Response

Please see the response in the attachment and the details in the manuscript.

Thank you

Reviewer 2 Report

It is a pleasure to be able to review this study and for that I would like to thank the editors for the opportunity to do so.
After reviewing the paper, I can conclude that the authors have done a good job in terms of writing the paper.
However, in order for the quality of the document to be adequate for the readers, I recommend the following modifications:

Revise the referencing standards used throughout the document. It seems that the authors have mixed two styles.
In the Background section the authors introduce interesting information about previous studies, but basic information about the object of study and the importance of carrying out this project is missing. Although a Theory section seems to be included, the information is disorganized and there is not much data to rely on.
I think it is important to reorganize the document and review the information included.
The graphic used in point 3. Research Framework, is very basic and I consider that it could be improved and a higher quality work could be done, according to the public to whom the information is destined.
As for the Discussion, this section is very brief and the results are not adequately discussed. I believe that more previous studies should be added and the data should be discussed more.
No sections on Limitations and Future Lines of Research and Practical Applications have been added.
The number of references used is very few. The bibliography should be increased in the sections throughout the document.

Author Response

(The authors gave the same response as above.)

Reviewer 3 Report

Thank you for sharing the manuscript “The Relationship between Emotional Intelligence and Pain Management Awareness among Nurses.”

The manuscript is interesting and easy to read. It is a little confusing.

Here are some suggestions:

The introduction could be shortened. The studies cited in this section could support the discussion to compare the current research results with what has already been published and explain the differences.

Please keep the quote consistent. Some references (introduction and discussion) are cited in a numbered format, and others in an APA format.

Table 4 does not seem necessary. It could be paired with table 3 or rewritten in one sentence.

A limitation section to discuss the validity of the measurement scales performed is essential.

I hope this will help.

Issam

Author Response

(The authors gave the same response as above.)

Reviewer 4 Report

Through this study, it was possible to know the importance of the nurse's emotional intelligence and pain management awareness. I am honored to read this paper as a reviewer, and would like to give some of comments for upgrading the first version of manuscript in order to make more readable to everyone.

1. Abstract

Please be specific about your results.

2. Introduction

 I am wondering this study is a really 'new attempt' than the approaches of existing studies, since it lacks persuasiveness that it can give new knowledge. Please clearly point out and explain what aspects are new compared to the previous studies.

3. Results

Please add footnotes to Table5

4. Discussion 

In results, it's necessary to verify sentences to help readers' understanding. It would be better to explain direction exactly. 

5. Reference

Please change Korean to English.

Author Response

(The authors gave the same response as above.)

Round 2

Reviewer 1 Report

The changes recommended were adequately addressed and the manuscript can be accepted.

Reviewer 2 Report

The authors have made the requested changes and the article can now be published.

Reviewer 3 Report

Thanks to the authors for making the necessary changes.

Reviewer 4 Report

You have logically and systematically increased the completeness of the thesis based on my advice. I hope that the results of your paper will be used in practice. 

This manuscript is a resubmission of an earlier submission. The following is a list of the peer review reports and author responses from that submission.